

# SINDRUM II

**Andries van der Schaaf**⋆

Physik-Institut der Universität Zürich, CH-8057 Zürich, Switzerland

⋆ andries@physik.uzh.ch

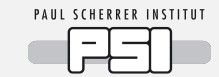

## Abstract

**In 1987 a collaboration including ETHZ - UZH - PSI - RWTH Aachen - Univ. Tbilisi proposed a new search for $\mu e$ conversion in muonic atoms. The SINDRUM II spectrometer came into operation in the $\mu$E1 area in 1989, but a dedicated beam line was delayed until 1998 by technical setbacks.**

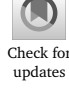
## 8.1 Introduction

$\mu e$-Conversion in muonic atoms would result in the emission of an electron with energy

$$E_{\mu e} = m_\mu c^2 - B_\mu - R_N, \tag{8.1}$$

with $B_\mu$ and $R_N$ being the muon binding energy and nuclear recoil energy, respectively. $E_{\mu e}$ is the endpoint energy of muon decay in orbit (MIO) where the energies of the two outgoing neutrinos vanish. For gold $E_{\mu e} = 95.55$ MeV [1]. Around the time of the SINDRUM II proposal, the best limit obtained for a heavy target was B($\mu^- $Pb$\to e^-$Pb) $< 4.9 \times 10^{-8}$ (90% C.L.)) [2].

## 8.2 SINDRUM II

To distinguish conversion electrons from MIO background at the planned sensitivity level, the spectrometer was designed with an energy resolution around 1% FWHM. SINDRUM II used a superconducting solenoid [3], formerly operated at the CERN ISR (see Figure 8.1). Two plastic scintillator hodoscopes (D) and a lucite Cerenkov hodoscope (E) are used for timing and triggering. The electron momentum is determined from the tracks recorded in the inner radial drift chamber (F), filled with $CO_2/iC_4H_{10}$ (70/30), a slow drift gas that results in a $6^o$ Lorentz deflection. The geometric acceptance for conversion electrons, when requiring the particle to completely cross drift chamber F before reaching an endcap detector, is 44% of $4\pi$ sr. The axial sense wires are located close to the outer cathode foil which is subdivided into

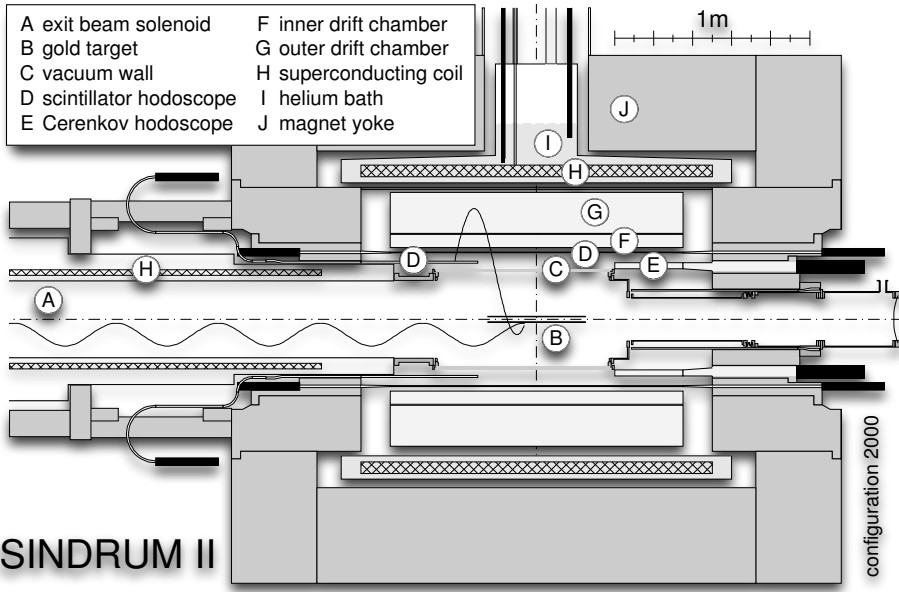

Figure 8.1: The SINDRUM II spectrometer as configured in the year 2000.

4.4 mm wide strips oriented $72^o$ relative to the wires. Correlated signals from wires and strips allow a 3d track reconstruction. The outer radial drift chamber (G) used a He/iC4H10 (88/12) gas mixture, that has a large radiation length to reduce multiple scattering. Figure 8.2 shows the online display of a multi-turn event recorded in 1989 with beam on. Note the energy loss along the spiralling path through the spectrometer. As can be seen in Figure 8.3, consecutive turns are always well separated so later tracks do not interfere with the first, main turn. The left side of the peak allows sensitive checks of the material budget and the momentum resolution.

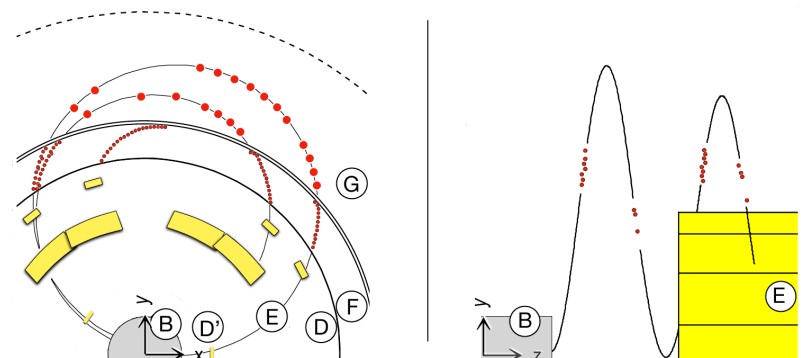

Figure 8.2: Traces of a 100 MeV/c $e^-$ in $xy$ and $zy$ views. The particle shown made 2 1/2 turns before leaving the tracker. Labeling is as in Figure 8.1.

## 8.3 The beam line

A beam pion stopping in the target produces isotropic background through radiative pion capture, followed by asymmetric internal and external $e^+e^-$ pair production, with a probability around $10^{-5}$ (see Section 8.5). Thus, no more than $10^4$ pions may reach the target during the entire data-taking period. Muons penetrate twice as deep into matter as pions of the

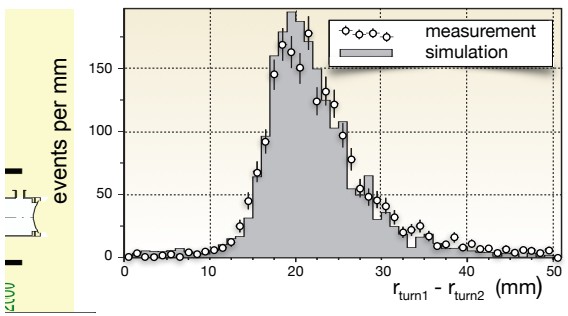

Figure 8.3: Change in radius of the first two turns of a multi-turn path caused by energy loss in the plastic hodoscope in particular. Thanks to this loss, the turns don't overlap, which otherwise might have confused the reconstruction.

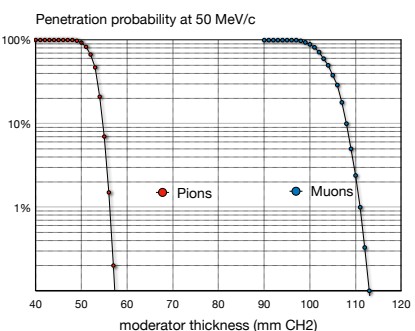

Figure 8.4: Simulated penetration probabilities of pions and muons in CH2 at 50 MeV/c.

same momentum (see Figure 8.4). This was utilized to eliminate beam pions: the fraction eliminated is limited by the high-momentum tail of the beam. The pion contamination was reduced in three steps (see Figure 8.5). First a momentum-selected beam was focused on a wedge-shaped degrader inside a final bending magnet. The few pions that penetrate do so with a wide momentum spread and have little chance to reach a second degrader in a collimator at the entrance of the transport solenoid. The beam was studied in great detail with dedicated diagnostic tools to tune the settings of the magnets and the slits. In this process the high-momentum tail of the beam was reduced by two orders of magnitude. Muons crossed the degraders but only very few pions emerged to enter the solenoid. These pions are slow and 99.99% decayed before reaching the target.

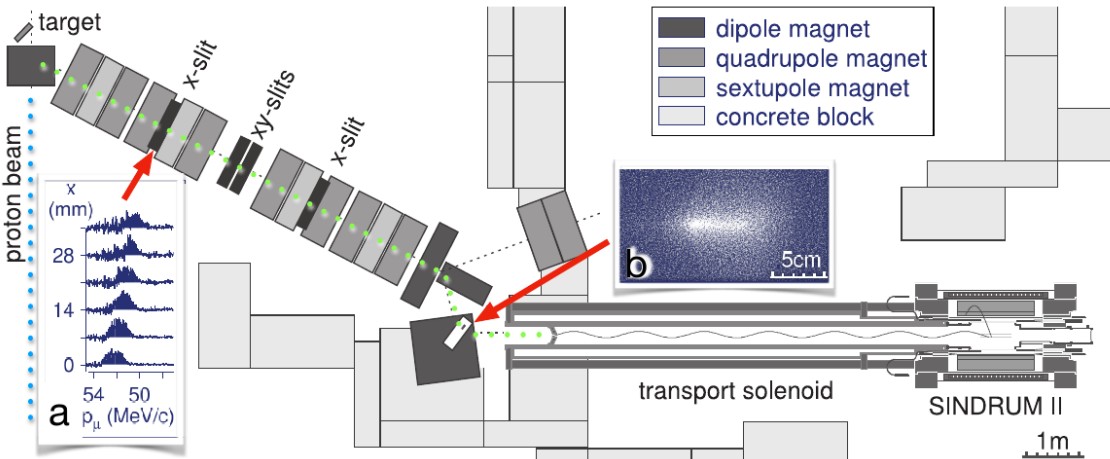

Figure 8.5: Plan view of the experiment at the $\pi E5$ secondary beam line during the final measuring period in the year 2000. A quadrupole channel extracted a beam with a similar amount of $\pi$'s and $\mu$'s in the backward direction from the production target. Inset **a** shows the impact of the momentum slit in the first dispersive focus. The momentum was determined by time of flight, based on the 50 MHz cyclotron rf signal. Inset **b** shows a CCD image of the beam spot. From here muons were guided to the target by a 9 m long transport solenoid.

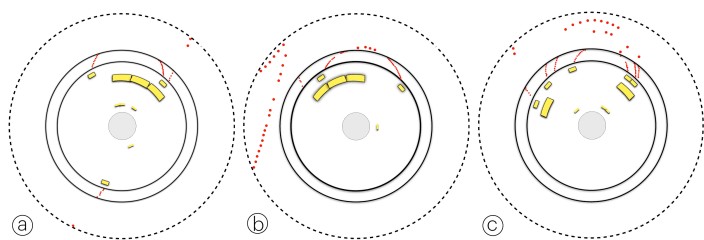

Figure 8.6: Three cosmic-ray events in the $r\phi$ projection. Signals recorded in the drift chambers (red), the plastic hodoscopes and Čerenkov counters (yellow) are indicated: a) a high momentum muon knocking an $e^-$ out of a Čerenkov counter, b) a high momentum muon creating an $e^+e^-$ pair in the magnet coil and c) an $e^+$ (most likely from the decay of a distant cosmic muon) spiraling in from outside.

Data was acquired even with the beam off as there are no beam counters in the final configuration. When requiring a circular track crossing drift chamber F, the trigger rate without beam was typically one per second. Figure 8.6 shows three examples.

## 8.4 Background

Cosmic-ray background was collected for more than a year with beam off: it can be recognized by the presence of additional signals in various detectors or by requiring the trajectory to originate in the target. What remains is associated with photons in cosmic-ray showers that enter through the cryogenic supply tower (see Figure 8.1). This background component was removed by an angular cut at the cost of a 5% loss in acceptance.

Another potential source of electrons with momenta around 100 MeV/c is radiative pion capture, mostly through intermediate photons producing asymmetric $e^+e^-$ pairs, in the target. Pion capture is much more likely in the moderator inside the collimator at the entrance of the transport solenoid (see Figure 8.5) and the resulting electrons and positrons may easily reach the target where they may scatter into the detector solid angle. This background can be recognized as it is strongly peaked in the forward direction and it has a characteristic time correlation with the cyclotron rf signal.

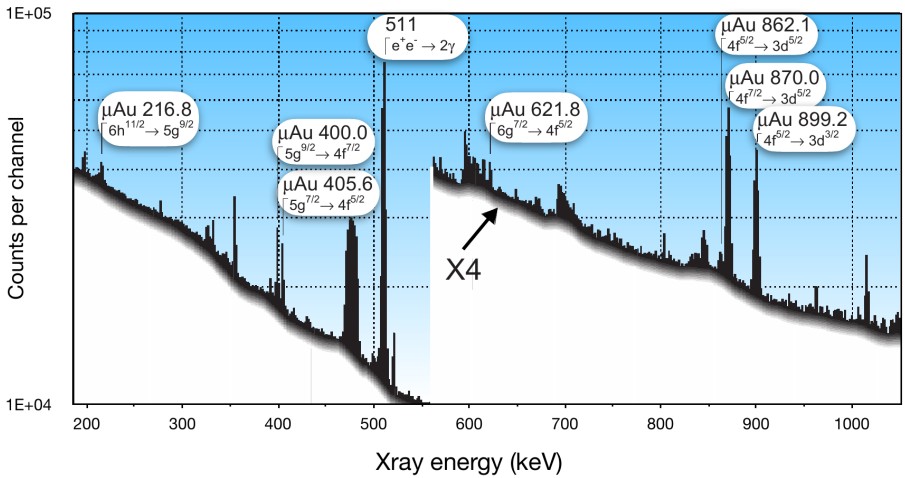

Figure 8.7: X ray spectrum recorded with a Ge(Li) detector during data taking to monitor the number of muons stopping in the gold target.

## 8.5 The 2000 data set

In the final 81-day period of data-taking in 2000 with a gold target,

$$N_{\mu stop} = (4.30 \pm 0.3_{stat} \pm 0.3_{sys}) \times 10^{13} \tag{8.2}$$

muons stopped in the target, as deduced from the muonic X-rays escaping the setup (see Figure 8.7). The monitor was calibrated with radioactive sources.

The analysis is based primarily on the momentum spectrum of electrons originating in the target. A cut is made on the position coordinates at the point of closest approach of the track to the central axis and is illustrated in Figure 8.8 for events surviving the cosmic-ray background checks.

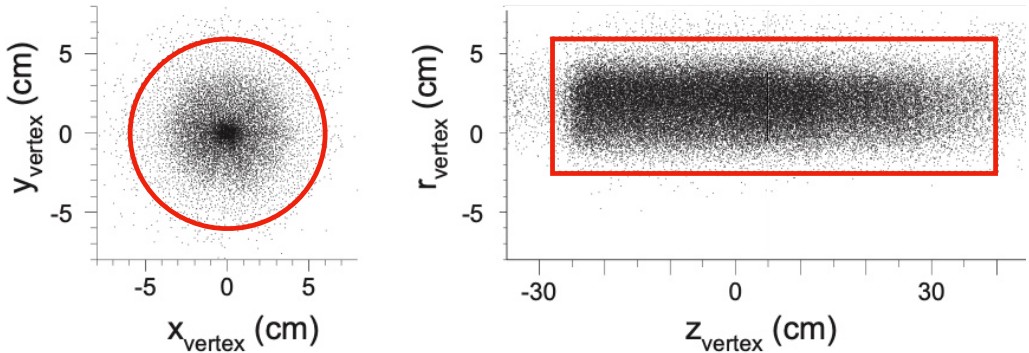

Figure 8.8: Reconstructed point of closest approach to the z axis in the *xy* and *zr* projections. The contours indicate the selected target region.

The vast majority of the selected events are muon decays in orbit (MIO). Following Shanker, the MIO spectrum used as input for the GEANT simulation has been approximated by [4]

$$N(E)dE \propto \left(\frac{E}{m_\mu c^2}\right)^2 \left(\frac{E_{\mu e} - E}{m_\mu c^2}\right)^5 dE + h.c. \tag{8.3}$$

The rate is proportional to $E^2$ at the low energy end, as is known from the Michel spectrum. At the high energy end, the rate falls proportional to the missing (neutrinos) energy to the fifth power. As shown in Figure 8.9 there is fair agreement between measurement and MIO simulation.

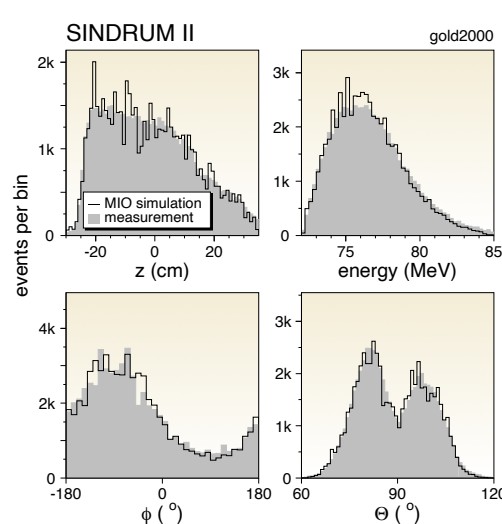

Figure 8.9:
Comparison of measurement and MIO simulation for four kinematic quantities.

The following comments may be helpful to explain some features:

- Muons come from $z<0$ and follow helical trajectories. Thus the stopping distribution falls from upstream to downstream.

- The fall of the rate at the low side of the energy distribution reflects the requirement that the electron crosses the inner drift chamber. This results in a transverse momentum threshold of around 70 MeV/c.

- There is a large $\phi$ anisotropy that is, however, antisymmetric about $0^0$, as expected for the up-down symmetry of the beam line (horizontal bending plane).

- The dip at $\theta = 90^o$ results from $e^-$'s that need too many turns to reach an endcap.

The $\theta$ and $\phi$ distortions are threshold effects that disappear towards $E_{\mu e}$.

The upper end of the electron momentum distribution, measured with a 53 MeV/c stopped $\mu^-$ beam, is compared with distributions from simulations of bound muon decay and coherent $\mu e$ conversion in Figure 8.10. The rate falls steeply towards $E_{\mu e}$ in agreement with the simulation, both in shape and in the number of events. Also shown are the results with 63 MeV/c stopped $\pi^-$ showing the enormous background reaching up to the pion mass, and the familiar Michel spectrum taken with 48 MeV/c $\mu^+$ beam. The $\mu^+$ data were taken at reduced spectrometer field for increased acceptance at the lower momenta and give an independent check of the momentum calibration and resolution.

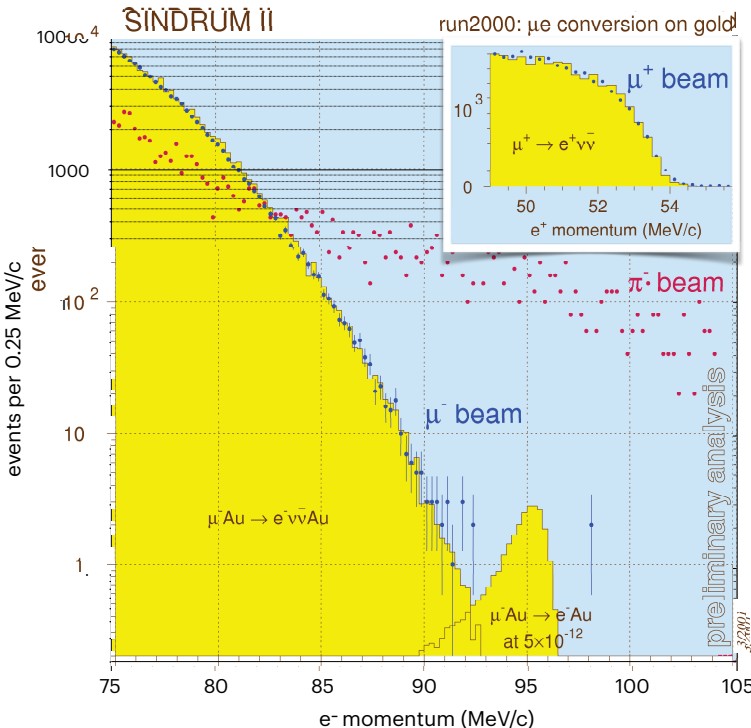

Figure 8.10: Momentum distributions for three different beam momenta and polarities: (i) 53 MeV/c negative muons, optimized for $\mu^-$ stops, (ii) 63 MeV/c negative pions, optimized for $\pi^-$ stops, and (iii) 48 MeV/c positive muons, optimized for $\mu^+$ stops. The 63 MeV/c data were normalized to the same measuring time. The measurement with the stopped $\mu^-$ beam is compared with GEANT simulations of decay in orbit and $\mu e$ conversion.

No convincing signal events are observed in the main $e^-$ momentum spectrum shown in Figure 8.10 and a maximum likelihood analysis of that spectrum results in a lowering of our own 90% C.L. upper limit by one and a half orders of magnitude. This result is included in Table 8.1 with all upper limits on $\mu^- e^-$ and $\mu^- e^+$ conversion obtained by SINDRUM II.

Table 8.1: SINDRUM II results over the years.

| beam line | year meas. | process | beam MeV/c | days | stops | upper limit 90 % C.L. | Ref. |
|---|---|---|---|---|---|---|---|
| | 1989 | $\mu^-$Ti$\to e^-$Ti | 100 | 25 | $4.28(32)\times10^{12}$ | $4.2\times10^{-12}$ | [5] |
| $\mu E1$ | 1992 | $\mu^-$Pb$\to e^-$Pb | 86 | 10 | $1.72(34)\times10^{12}$ | $4.6\times10^{-11}$ | [6] |
| | 1993 | $\mu^-$Ti$\to e^+$Ca | 86 | 60 | $2.76(21)\times10^{13}$ | $7.3\times10^{-13}$ | [7] |
| $\pi E5$ | 1997 | $\mu^-$Au$\to e^-$Au | 20 | 24 | $7.6\times10^{11}$ | $1.91\times10^{-11}$ | [8] |
| | 2000 | $\mu^-$Au$\to e^-$Au | 53 | 81 | $4.37(32)\times10^{13}$ | $7\times10^{-13}$ | [9] |

## 8.6 Conclusions and outlook

After a decade long campaign, SINDRUM II took its final data in 2000. The resulting upper limits on $\mu e$ conversion were pushed below $10^{-12}$. The effort took longer and brought us not quite as far as was promised in the proposal but now, almost twenty years later, the SINDRUM limits still stand. The new more ambitious experiments are simply getting bigger, more complex, more expensive, require more manpower and often rely on new detector concepts and thus time consuming R&D.

There are two new efforts planning to continue where SINDRUM II ended: COMET (J-PARC, Japan) [10] and MU2E (Fermilab, U.S.A) [11]. Both use a pulsed beam and a delayed time window to fight prompt (pion) background which excludes heavy targets such as gold, with their correspondingly short decay times. Both use a staged approach, so with a bit of luck, new territory may be reached before the end of the decade.

The "search for nothing" keeps moving on!

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
