# Peer review of "SINDRUM II"

_SciPost Physics Proceedings, doi:SciPost Phys. Proc. 5, 008 (2021)_

## Round 1 · Referee Report · Anonymous (Referee 1) · 2021-4-12

Strengths

1) Paper very clearly and concisely describes the detector and the relevant beam components. 2) All relevant details to understand the experiment are given. 3) A clear overview of all results obtained by SINDRUM-II is quoted.

Weaknesses

1) Paper could gain by adding just a few short sentences on the motivation and put this experiment into context. 2) Mention, what was the status of mu-e conversion at the onset of SINDRUM-II? 3) What was the limiting factor of SINDRUM-II? What were the main systematics?

Report

The SINDRUM-II was clearly an important experiment among all the rare muon experiments. The paper very concisely describes the techniques used for the measurements and lists all obtained results. It is well and understandably written; thus the text does not need changes. The results obtained, i.e. upper limits, are of high relevance in the context, and are still valid as of today. Therefore, the paper certainly should be published, as it constitutes one of the very important contributions to the review of Particle Physics at PSI.

The paper could gain by adding a few lines on I) the motivation for the measurement, ii) the context of SINDRUM-II at the time of "start", and on iii) the limitations of the experiment.
Also, it could be helpful to know, what was or what caused the substantial delays of the beam mentioned in the text, and if there were several different beam settings or stages.

Requested changes

1) add something about the status of the field at the start of the experiment SINDRUM-II (earlier experiments, limits...?). 2) Increase the size of the figures (in particular of 8.1, 8.2., 8.5). There is enough space in width available. 3) line 35: Background produced by pions: what background and if in particular directions or locations? everywhere in the detector? 4) Fig.8.5: is this the beam in its final setting? 5) line 59: where exactly was the moderator located? can it be indicated in fig. 8.5? 6) line 95: Which distribution was used in the analysis to determine the number of observed events as none?

---

## Round 1 · Referee Report · Adrian Signer (Referee 2) · 2021-4-22

Report

We (the editors Cy Hoffman, Klaus Kirch, Adrian Signer) had the
opportunity to review an earlier draft of the article and were in
communication with the author before the submission. All our comments
and suggestions have been taken into account. Hence, we think the
paper can now be published in the current form.

---

## Round 2 · Author Response

Revised version following suggestions by the referee's

---

## Round 2 · List of Changes

- included result existing by the time of the proposal
- increased size of some figures
- elaborated on background produced by pion stops upstream of the target
- included information in caption of Fig.8.5
- included information on the location of second moderator
- mentioned maximum likelihood analysis

---

## Editorial Decision

published